# Measuring Relative Component Motion and Stability in Total Hip Replacements Using a Magnetic Position and Orientation Sensing System [note 1]

**DOI:** 10.3390/s25237280

**Published:** 2025-11-29

**Authors:** Oliver G. Vickers, Peter R. Culmer, Graham H. Isaac, Robert W. Kay, Matthew P. Shuttleworth, Tim N. Board, Sophie Williams

**Affiliations:** 1The Faculty of Engineering and Physical Sciences, University of Leeds, Leeds LS2 9JT, UK; 2The Centre for Hip Surgery, Wrightington Hospital, Wigan WN6 9EP, UK

**Keywords:** total hip replacement, instrumented orthopaedic prosthesis, magnetic position and orientation sensing system, joint impingement, subluxation

## Abstract

An instrumented total hip replacement (THR) implant capable of remote and continuous monitoring would be an attractive prospect for a surgeon to conveniently track the recovery of their patients. Measuring the relative motion of the prosthesis components would provide insight into joint kinematics and contribute to the detection of adverse events including impingement and subluxation. The aim of this study was to develop a sensing system to measure the relative orientation and translation of the prosthesis components. A tri-axis magnetometer and a permanent magnet were integrated into clinically available THR components, forming a magnetic position and orientation sensing system. A robotic arm was used to articulate the components through controlled motion routines and record the orientation of the components. The output of the robot arm and a camera tracking system were used to validate the performance of the sensing system. The sensing system measured the relative orientation of the components to two degrees of freedom with an RMSE of <4.0° and measured the displacement of the femoral head during an impingement-driven subluxation motion with an RMSE of 0.2 mm. This proof-of-concept work has shown that magnetic sensing technology can track the position and orientation of THR components. With further development, this sensing method could feature within an instrumented THR implant.

## 1. Introduction

Total hip replacement (THR) is a widely used intervention for late-stage osteoarthritis. In 2023, over 100,000 THR surgeries were performed in England, Wales, and N. Ireland [1]. The frequency of THR surgery is increasing along with the need for implants to be reliable, have good survivorship, be cost effective, and enable surgeons to provide excellent outcomes in all cases to reduce the growing burden on clinicians, healthcare providers, and the economy.

A THR aims to return function back to the patient’s hip by restoring the joint’s natural centre of rotation, weight bearing capability, and ball-and-socket mobility. During THR surgery, the body’s damaged tissue is replaced with artificial components. A THR consists of a metal stem positioned in the central canal of the femur with an affixed femoral head, i.e., the ball. A metal shell component is fixed within the acetabulum of the pelvis and, typically, a liner made from a high-wear-resistant material (e.g., polyethylene) is positioned within the acetabular shell, thus replicating a socket. A diagram of a THR implanted within a human hip is shown in Figure 1.

Instrumented implants capable of in vivo sensing would allow for the continuous monitoring of patients post-operatively. Data collected from the implant could provide insight into the function of the implant in vivo and track the recovery of the patients [2,3]. The recent release of the Persona IQ^®^ The Smart Knee^®^ (Zimmer Biomet, Warsaw, IN, USA) replacement system has shown that a sensorised prosthesis for regular clinical use is feasible from a technical, regulatory, and business perspective [3,4]. However, to date, integrating sensing systems into THRs has focused on measuring joint reaction forces [5,6] and component loosening [7,8,9]. These systems were prototypes or only intended for research purposes. Tracking joint kinematics would allow for the collection of metrics such as joint range of motion and step count, which are commonly used to monitor post-operative recovery. Furthermore, measuring the components relative position and orientation would facilitate the detection of adverse events, for example, impingement and subluxation, and so give early indications of component damage or the need for clinical intervention.

Permanent magnets coupled with a magnetic field sensor can provide a contact-free method for position, proximity, and angle measurement. Magnetic position and orientation (MPO) systems have been used previously to measure linear displacement [10], one-degree-of-freedom (DoF) orientation [11], two-DoF orientation [12], three-DoF orientation [13,14,15], and a three-DoF tactile force sensor [16]. Arami et al. [17] developed an instrumented knee implant which featured three three-axis anisotropic magnetoresistance sensors embedded within the polyethylene insert and a permanent magnet positioned in the femoral component of the knee replacement. They used a neural network to map the sensor-recorded magnetic field data to 15 orientation states of combined flexion/extension and abduction/adduction rotation (sagittal plane angle range from 138° to 192° and coronal angle range from −8° to 4°). The system had a reported RMSE of 2.35° and 0.31° in flexion/extension and abduction/adduction, respectively. The use of an MPO sensing system as a method of providing contact-free position and angle measurement of THR components has not been explored.

Work lead by Michael Ortner including the publications [13,14,15] has shown that orientation tracking to three DoF can be achieved with a single 3-axis magnetic field sensor MPO sensing system. The application described in those studies was a custom-made joystick or multimedia control element; therefore, the design constraints were less rigid and the required sensing range and resolution was less than that required for the present application. A design specification of the present work dictated that there should be no alterations to the THR component geometries. Consequently, the method used by Ortner and colleagues to achieve orientation tracking to three degrees of freedom could not be replicated as it would have required positioning the magnet or sensor away from the central axis of the stem, i.e., embedded within the body of the femoral head or polyethylene liner. Therefore, an alternate method described by Lutz and Folletto [12], which can derive orientation to two DoF, was considered. Measuring the relative orientation of the THR components to two DoF would still provide clinical utility and was an intuitive method of visualising and describing the location of a component–component impingement point.

The aim of this study was to develop and integrate a magnetic position and orientation sensing system into a THR to measure the relative motion between the components and validate a proof-of-concept prototype in terms of measuring the femoral stem orientation to two DoF and detecting THR bearing surface separation during impingement-driven subluxation.

## 2. Materials and Methods

This methods section is separated into three parts. Firstly, the development of an MPO system for use in THRs to measure orientation to two DoF is considered. This includes details of how orientation to two DoF was defined, how the magnetic field sensor and magnet were integrated into the THR components, and how the size and shape of the magnet were selected. This section is concluded with descriptions of three methods for mapping the magnetic field readings read by the sensor to the position and orientation of the magnet and the protocol used to calibrate the magnetometer. The second section describes the experimental method used to validate the performance of the developed MPO and, finally, the third section presents a method for using the developed MPO to measure bearing surface separation and how that was validated experimentally.

### 2.1. Development of a Magnetic Position and Orientation Sensing System for Use in a Total Hip Replacement to Measure Orientation to Two DoF

A THR operates as a ball-and-socket joint to replicate the biomechanics of a human hip joint. In this study, the orientation of the THR components (i.e., the angle of the femoral stem neck with respect to the acetabular liner) was described to two DoF using tilt and azimuth angle. Tilt angle (Figure 1b) is the angle that the central axis of the stem’s neck rotates from the central axis of the acetabular components, and azimuth angle (Figure 1c) is the angle that the central axis of the stem’s neck rotates about the central axis of the acetabular components. The required sensing range was from 0° to 65° in the tilt angle direction and from 0° to 359° in the azimuth angle direction, as this was the maximum range of motion of the components. The required sensing resolution was set at 1° as this was deemed to be a clinically useful resolution level.

#### 2.1.1. Integrating Magnet and Sensor into Total Hip Replacement Components

A three-axis magnetometer (mlx90393, Melexis, Ypres, Belgium) on a custom-made PCB board was potted into the introducer hole of a clinically available acetabular shell (56 mm outer diameter). A spigot stem component was 3D-printed (PLA) and matched the geometry of a clinically available stem taper and included a pocket on the face of the taper to hold a neodymium (N42) disc (8 mm × 2 mm) magnet (Magnet Experts Ltd., Tuxford, UK). The optimisation process used to select the 8 mm × 2 mm disc magnet is explained in the following section. A 36 mm metal femoral head was pressed onto the stem taper and a polyethylene liner was pressed into the acetabular shell. A cross-section diagram shows the magnet and sensor positioned in the assembled THR components (Figure 1d).

#### 2.1.2. Optimisation of Magnet Size and Shape for Orientation Tracking to Two DoF

The sensitivity of an MPO sensing system is dependent on the step size between each state (which is a chosen system parameter) and the state separation (the extent to which the magnetic field values within the dataset are different). A system with a large state separation will be more adept at detecting changes in the position or orientation state of the magnet [15]. Malago et al. [15] evaluated the minimum state separation of an MPO system as a means of providing an estimation of the quality of a specific MPO implementation. They used this quality factor as a quantitative measure to guide the selection of the system parameters to optimise the performance of their MPO system.

Simulations of a magnet’s motion and magnetic field were produced based on the defined physical parameters of the system using Magpylib © Python package (Version: 4.0.0b2, Author: Michael Ortner). The library includes methods for translating and rotating magnet features and a sensor class that uses analytical equations to return the magnetic field vector at a specified location. The output of the simulation was a dataset of magnetic field values at the location of the sensor ($B_{n}$) at every orientation state (${States}_{n}$), i.e., tilt angle from 0° to 65° in steps of 1° and azimuthal angles from 0° to 359° in steps of 1°. The mean and minimum state separation of ($B_{n}$) was evaluated using the scipy.spatial.KDTree nearest neighbour algorithm [18], which is the same as that described by Maneewongvatana and Mount [19].

This procedure was repeated nine times for nine different simulated magnet shapes and sizes. The array of magnets investigated were all able to fit within the maximum magnet space (approx. 12 × 3 mm cylindrical region) and were selected from the stock of a supplier (Magnet Experts Ltd., Tuxford, UK). Additionally, three hypothetical (hyp) magnets were also added to the array that were of conceivable shape and magnetisation direction. The range of magnets was selected to cover a variety of shapes, sizes, and directions of the magnetisation vector.

#### 2.1.3. Implementation of Numerical Methods to Map Magnetic Field Data to the Orientation and Position of a Permanent Magnet

Three unique tracking methods were investigated including look-up table, neural network, and responsivity tracking. Each was chosen as a method of mapping magnetic field data (produced from simulations or gathered experimentally) to the position and orientation of the magnet and, thus, the position and orientation of the THR components.

##### Look-Up Table

A look-up table is the simplest type of tracking method explored in this work. Lumetti et al. [14] used a look-up table as an inversion method for their three-DoF multimedia control joystick, which is a similar application to the present work. The computational efficiency of a look-up table will progressively decrease as the size of the table increases. This is due to the speed it takes to run the algorithm (i.e., search the dataset) and the memory space required to store the reference dataset *B**n*. The issue of operation speed and memory space may become more apparent with the addition of more DoF.

The look-up table method was realised using a custom Matlab (Matlab R2023a, Mathworks, Natick, MA, USA) function. The function takes in the sensed magnetic field value and subtracts it from all the magnetic field values in the reference *B**n* dataset, thus returning *B**d**i**f**f* (which is the same length as the *B**n* and *S**t**a**t**e**s**n* arrays). The index of the minimum difference is then found and used to return to the corresponding position and orientation state stored in the *S**t**a**t**e**s**n* array.

##### Neural Network

Fitting the data using a neural network produced a function to relate magnetic field data B_n to the desired position and orientation state, e.g., the tilt angle [*θ*] and azimuthal angle [*ψ*]. For tracking the neck of the stem orientation to two DoF, the Matlab (Matlab R2023a, Mathworks, Natick, MA, USA) NNFit toolbox was used to train the neural network using a Levenberg–Marquardt backpropagation algorithm for training [20]. The deep neural network structure included three hidden layers with 12, 12, and 12 neurons, respectively, and outputted tilt and azimuth angles. The dataset was randomly divided amongst training 70% (samples), validation 15% (samples), and testing 15% (samples). This implementation of a neural network is similar to that used by Jones et al. [16], who developed a magnetic-sensor-enabled soft tactile force sensor. The chosen settings matched those suggested as standard by the Matlab (Matlab R2023a, Mathworks, Natick, MA, USA) NNFit toolbox.

##### Responsivity Tracking

As the stem’s neck is tilted away from the central axis, the magnet is translated in the XY plane, which increases the sensed signal in the XY plane and correspondingly decreases the sensed signal in the Z direction. This relationship of magnetic field responsivity to magnet tilt angle has been used by Lutz and Foletto [12] to track the tilt angle of a joystick. They expressed responsivity as a ratio to mitigate the impact of changes in the distance between the magnet and the sensor (i.e., vertical play of the stem) over the lifetime of the joystick. They calculated their ratio using (1):(1)$$R_{ratio}=\sqrt{{\left(\frac{B_{x}}{B_{z}}\right)}^{2}+{\left(\frac{B_{y}}{B_{z}}\right)}^{2}}$$
where $B_{x}$, $B_{y}$, and $B_{z}$ are the magnetic fields measure in the X, Y, and Z directions, respectively. An alternate method of calculating a responsivity ratio (a modification to the method described by Lutz and Foletto) was used in this work. The responsivity term is calculated using (2):(2)$$R_{ratioBMagnitude}=\sqrt{{\left(\frac{B_{x}}{\left|B\right|}\right)}^{2}+{\left(\frac{B_{y}}{\left|B\right|}\right)}^{2}}$$
where $\left|B\right|$ is the magnetic field vector. The **R_ratioBMagnitude_** value, evaluated (using simulated data) over the range of stem tilt angles from 0° to 66° (maximum tilt angle of the stem component), was found to have a more linear response for the present system when compared to the $R_{ratio}$ used by Lutz and Foletto. The simulated **R_ratioBMagnitude_** values were stored in a look-up table with the corresponding tilt angle value.

The inverse tangent function has been suggested by Melexis for tracking the rotary position and heading of a magnet in a position sensor application [11]. The projection of the magnetic field vector on the XY sensing plane reflects the azimuthal angle of the joystick. Therefore, the azimuthal angle can be calculated using (3):(3)$$Azimuthal\;Angle(\psi )=arctan(B_{y},B_{x})$$
where, if the neck of the stem azimuthal angle is 0°, it is aligned with the positive *x*-axis and, when it is 90°, it is aligned with the positive *y*-axis. Using the responsivity ratio and trigonometric function is advantageous as it reduces memory on an embedded system and is computationally faster compared to look-up table functions.

#### 2.1.4. Magnetometer Calibration

The magnetometer boards were wired to a Teensy 4.0 microcontroller (PJRC, Sherwood, OR, USA) and the sensor data were read in via an I2C serial connection. A dedicated library (Adafruit_MLX90393 library, Adadruit Industries, Brooklyn, NY, USA) was used to configure the sensors. This resulted in the sensor reporting magnetic field values at a measured frequency of 49 Hz and at an approximate sensitivity of 6.01 µT in the X and Y direction and 9.68 µT in the Z direction. A calibration algorithm described by [21] and implemented using the magcal function (Matlab R2023a, Mathworks, Natick, MA, USA), which was applied to the raw magnetic field readings to counteract the hard-iron and soft-iron interferences that typically interfere with magnetoresistive sensor readings.

### 2.2. Experimental Setup for Verification of Tracking Method Performance for Orientation Tracking to Two Degrees of Freedom

A UR3 robotic arm (Universal Robots, Odense, Denmark) was used to simultaneously manipulate the components and record the position and orientation of the femoral head as a ground truth measure. The robot arm was mounted on a custom aluminium extrusion base frame to ensure it was affixed to the table and would keep the robot arm stable during operation. The spigot neck feature was extended and included a flat base to allow for secure mounting to the robot tool flange. The femoral head component was then pressed onto the stem fully so that the top face of the magnet was coincident to the head taper face. The acetabular shell component was held in a 3D-printed (PLA) mount with a hemispherical recess the same dimension as the outer diameter of the shell component (56 mm). An acyclic sheet with a through hole smaller than the diameter of the face of the shell component but larger than the diameter of the face of the liner component was screwed in place on top of the mount and used to clamp the shell and liner into the mount. An image of the experimental setup is shown in Figure 2a. The head was positioned manually in the liner with the robot in Freedrive mode. Once the femoral head was located into the liner, the robot was then commanded to align the stem vertically (and so perpendicular to the face of the shell and liner) whilst keeping its position constant, ensuring the femoral head remained located in the liner. This was defined as the stem in the neutral position and was recorded onto the robot controller to keep component positioning consistent throughout testing. The robot arm was programmed to rotate the stem whilst the femoral head was located in the liner component, thus replicating the motion of a ball-and-socket joint. To test the performance of the MPO system measuring stem orientation to two DoF, the robot manipulated the stem from the neutral/vertical position to a tilted position and then rotated the stem through a full sweep of azimuthal angles from 0° to 355°. The azimuthal sweep was repeated for a range of tilt angles from 5° to 60° with steps of 5°. The experimental run was performed three times for each of the tracking methods (look-up table, neural network, and responsivity tracking).

The output of the MPO sensing system was recorded throughout the motion of the magnet and the robot arm recorded the femoral head position at their maximum sampling rates of 50 Hz and 125 Hz, respectively. At the start of each test run, the robot sent a trigger signal to a microcontroller, which meant the data from the two sources could be synched. To obtain datasets of the same size for statistical analysis, timestamps of the sensor data were matched to the closest timestamp of the robot data. The remaining robot data were omitted, leaving datasets of the same size.

To compare the sensing system tracking methods, the error between the robot-reported orientation (expected) and the sensing-system-predicted orientation (observed) was computed for each angle direction using (4). The sine and inverse sine functions were used to remove the effect of crossing the azimuth angle singularity 360°/0°. The RMSE (5) ± standard deviation (SD), minimum error, maximum error, and range of error were computed for all the data across the three repeats.(4)$$Error=\sin^{-1}(\sin(Observed\;Value))-\sin^{-1}(\sin(Expected\;Value))$$(5)$$RMSE=\sqrt{mean({Error}^{2})}$$

### 2.3. Measuring Bearing Contact Surface Separation During an Impingement-Driven Subluxation Event

An impingement-driven subluxation is created when the femoral head is levered out of the liner component with the component–component impingement point, acting as the fulcrum of the lever (Figure 2b,c). Subluxation can lead to liner damage, cause patient pain and discomfort, and be a precursor to dislocation [22,23]. An impingement-driven subluxation event was recreated using the robot arm to rotate the stem to an extreme angle beyond the normal working range of motion of the components, thus causing the components to impinge and the femoral head to lever out of the liner. A camera tracking system (CTS) consisting of a digital camera (Panasonic Lumix DMC-GF6K Panasonic, Osaka, Japan) and the Matlab (Matlab R2023a, Mathworks, Natick, MA, USA) Image Processing and Computer Vision toolbox was used to track the location of two colour markers that were positioned on the stem component. The position of the markers was then used to infer the location of the centre of the femoral head during the impingement-driven subluxation motion.

Copper foil was wrapped around the 3D-printed stem component and positioned around the region of the liner’s rim where the stem would contact if component impingement occurred, as shown in Figure 2a. This formed a continuity switch where contact between the copper foil on the stem and on the rim of the liner completed the electrical circuit. The continuity switch was used and recorded in the robot program to indicate that impingement contact had occurred and, therefore, when an impingement-driven subluxation event began and ended. The impingement-driven subluxation motion was performed with the stem being rotated to a maximum angle of 74°. The (CTS) recorded the position of the centre of the femoral head and the position and orientation of the reference magnet. These position and orientation data were inputted into the Magpylib © simulations to generate magnetic field data for a look-up table. Twenty-five values were recorded during the subluxation event and used to populate the look-up table. The range of positions in the Z direction prediction was from 0 mm to 3.14 mm, and the mean step size between values of effective resolution was 0.15 mm.

The impingement-driven subluxation motion test protocol began with the robot arm rotating the stem until the stem and liner components contacted (Figure 2b). The continuity switch was used to detect the component contact and triggered the robot arm to return. This recreated an impingement event but no bearing surface separation, so not an impingement-driven subluxation event. The stem was then rotated to three angles beyond the normal working range of the components (beyond a tilt angle of 64°), namely, 70°, 72°, and 74°. These angles were chosen as they created three distinct impingement-driven subluxation events with evenly distributed levels of femoral head displacement.

The recordings of the motion protocol were manually inspected frame by frame and the researcher judged the first occurrence of the femoral head moving out of the liner, which indicated the start of the impingement-driven subluxation. The camera recorded at a frame rate of 25 frames/s, which resulted in the CTS recording position and orientation at 25 Hz. To obtain datasets of the same size for statistical analysis, timestamps of the extracted CTS coordinate data were matched to the closest timestamps of the sensor and robot data. The remaining sensor and robot data were omitted, leaving datasets of the same size.

## 3. Results

### 3.1. Results from Optimisation Study That Informed Magnet Selection

The mean and minimum state separation (meanSS and minSS) values for the nine magnets investigated in the simulated optimisation study are shown in Table 1. The 8 × 2 mm disc magnet returned the greatest meanSS of 10.4 µT and minSS of 0.522 µT values. This magnet had the largest volume of the investigated magnets at 100.5 mm^3^ and the magnetised face of the magnet points towards the sensor, resulting in the maximum magnetic field value measured by the sensor of 1996.8 µT, which was the greatest out of all the other investigated magnets. Therefore, the 8 × 2 mm disc magnet was deemed the more optimal for this application and it was expected that it would result in the best sensing performance.

A plot of the magnetic field vector state separation values when tilt angle increases by 1°, starting at 0° for the 8 × 2 mm disc magnet configuration, is shown in Figure 3. The state separation when only the azimuthal angle state changes from 0° to 1° at every tilt angle state is also shown in Figure 3. At every tilt angle state, the state separation was constant across the full azimuthal angle range (from 0° to 359°). Overall, the change in tilt angle resulted in a greater state separation than change in the azimuth angle direction. Furthermore, when the tilt angle was <20°, the change in the azimuth angle had a state separation that was less than the sensitivity of the magnetic field sensor (9.68 µT). This suggests that there will be a higher level of error when the stem is in the low tilt angle orientation (<20°) as the sensing system will not be able to distinguish between the azimuth angle states.

### 3.2. Measuring Hip Replacement Component Orientation to Two Degrees of Freedom

The RMSE ± SD and minimum, maximum, and range of the error between the robot arm stem angle versus tracking-method-predicted angle over the three repeats of the three tracking methods are shown in Table 2. Error in the azimuthal angle was greater than in the tilt angle direction. Error in the azimuthal angle was greater in the low-tilt-angle region, i.e., <20°. Error in the tilt angle direction was greatest in low-tilt-angle region, i.e., <20°, and in the high-tilt-angle region, i.e., >45°. The performance of the responsivity tracking method and the look-up table method are comparable and, in general, performed better than the neural network method.

### 3.3. Measuring Bearing Surface Separation of Total Hip Replacement Components During an Impingement-Driven Subluxation Event

The MPO sensing system was able to distinguish between the different severity levels of an impingement-driven subluxation event and not report a false negative only when an impingement event occurred with no bearing surface separation. The plot of the CTS-recorded position and MPO-sensing-system-predicted position of the centre of the femoral head in the *z*-axis with respect to the joint centre of rotation is shown in Figure 4. The data shown are for all three repeats of the impingement-driven subluxation protocol in turn. The RMSE (±standard deviation) of the reported position over the full dataset for the CTS versus the MPO sensing system was 0.2 mm ± 0.2 mm.

## 4. Discussion

This novel magnetic position and orientation sensing system was able to measure the relative motions between the components and so derive the femoral stem’s neck orientation to two DoF and THR bearing surface separation during impingement-driven subluxation. The sensing system uses small form factor components and does not require contact with or alteration to the functional bearing surfaces.

The most comparable study to the present work is that published by Arami et al. [17,24,25]. They reported achieving an RMSE in flexion/extension and abduction/adduction of 2.35° and 0.31° over the angle ranges from 138° to 192° and from −8° to 4°, respectively. The difference in joint replacement being considered in Aramis’s study (total knee replacement) and the present work (total hip replacement) means a direct comparison of the results cannot be made. However, tilt angle, as described in this work, is the most comparable angle to the flexion/extension and abduction/adduction angles used in Arami’s work. The accuracy in the tilt angle direction achieved in this work (RMSE 1.0°) is an improvement on that achieved by Arami in flexion/extension but not abduction/adduction. There is scope for the methods described in the present study to be translated into a sensing system for use in a TKR prosthesis. Arami et al. were measuring over a smaller sensing range; therefore, there could be scope that, with additional tuning and optimisation, a similar level of accuracy to that reported by Arami et al. in abduction/adduction angle could be possible.

Presently, the sensing system only reports orientation to two DoF. However, hip joint rotation is typically described to three DoF as rotations in the planes of the body, namely sagittal plane flexion/extension (FE), coronal plane abduction/adduction (AA), and transverse plane internal and external rotation (IE). Therefore, measuring orientation to three DoF would bring more clinical utility to the sensing system. Future work can explore achieving orientation tracking to three DoF by reconsidering the method described by Ortner and colleagues [13,14,15]. This would require reconsidering the initial design specification to not make alterations to the THR component geometries and instead have the magnet or sensor embedded within the body of the femoral head or liner. However, this may result in poorer sensing performance (range, accuracy, and resolution) and may require additional magnetic field sensors.

Using the tilt and azimuth angle orientation notation created singularities in the azimuth angle, i.e., when the stem was in the neutral position and, secondly, when the stem crossed the azimuth angle value of 0°/360°. The results of the optimisation study showed that, when the stem is at a tilt angle of <20°, the sensitivity of the sensor is less than the state separation between the neighbouring azimuth angle states. These factors all contributed to high levels of error when the stem was in these regions. The optimisation of the spacing of the orientation states included in the sensing range could improve sensing performance, for example, by increasing the azimuth angle step size to >1° in the low-tilt-angle region (<20°) or evenly spacing the discrete orientation states by using an algorithm like that of the Fibonacci sphere algorithm. Overall sensing performance could be improved by configuring the magnetic field sensor to improve the resolution and sampling rate. More advanced look-up table methods could be investigated to improve performance, for example, curve fitting to the responsivity ratio values or the use of a parametric equation of a surface fitted to the *B**n* dataset. The comparatively poor performance of the neural network method suggests the model is not fully optimised. The training dataset size and the number and size of network layers were not altered and there was no examination into whether the model was over- or under-fitted. Future work can explore addressing these factors with the aim of improving the performance of the neural network, thus reducing the error of the orientation prediction.

When simulating an impingement-driven subluxation event, the robot arm did not apply force axially through the stem. This meant that the femoral head was not pushed up the far side of the cup of the liner, which typically happens during an impingement-driven subluxation event [26]. Therefore, a method of applying anatomically relevant forces through the implant components is needed to simulate a more biomechanically accurate impingement-driven subluxation event. This will likely be achieved by using validated in vitro experimental hip simulators [27] or using a more capable robotic arm like that used by [28,29]. Furthermore, when considering the physiological environment, the system will need to account for long-term changes in the position of the components, for example, because of implant precession, osseointegration of the prosthesis, and the evolution of the bone material around the implant, which will affect the achievable accuracy of the system.

An alternate method of integrating the sensing system components into the THR would be needed for in vivo use. However, before that can be considered, the integration of a magnet into an implanted THR may be hazardous to the patient if the patient comes into proximity with a high-powered magnet, e.g., an MRI scanner. Future work would need to involve computational and experimental studies investigating the safety of implanting a prosthesis with an embedded permanent magnet.

## 5. Conclusions

This work is the first to show that a magnetic position and orientation sensing system can be used within a THR to measure the kinematics of the implant components. The output of the sensing system can measure the orientation and position of the components to a clinically relevant level of accuracy (RMSE 1.0° and 4.0° in the tilt azimuth angle directions) and detect the occurrence of adverse events unique to THRs, i.e., impingement and impingement-driven subluxation.

This is a proof-of-concept prototype that is not validated or ready for use in an in vivo sensing-enabled THR prosthesis. To realise this, considerations will need to be made on safety, better integration into the implant for biocompatibility, power supply, and wireless communication. Future work will also need to include improving the functionality, for example, by extending the sensing capabilities of the system to capture the full three DoF of the stem and to extracting metrics to characterise common activities of daily living or adverse events that might occur within a THR, e.g., steps, cadence, unusual gait cycle pattern, or subluxation.

## Figures and Tables

**Figure 1 sensors-25-07280-f001:**
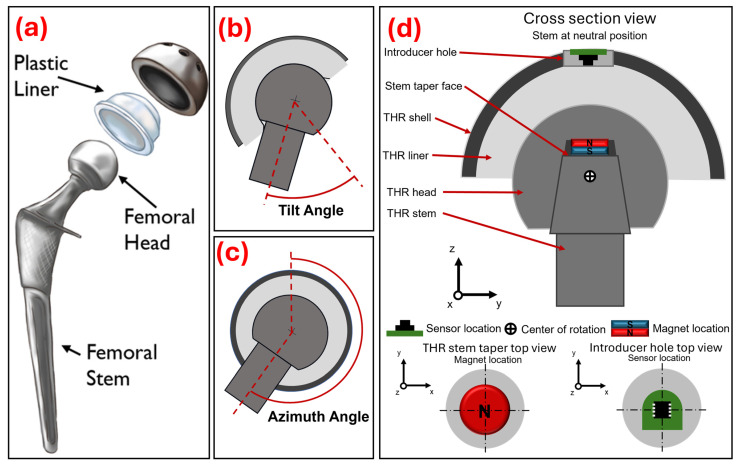
(**a**) Exploded diagram of typical hip replacement components reproduced with permission from OrthoInfo. © American Academy of Orthopaedic Surgeons. https://orthoinfo.org/ (accessed on 25 November 2025), (**b**) Diagram representing femoral stem tilt angle with respect to the acetabular components. (**c**) Diagram representing femoral stem azimuth angle with respect to the acetabular components. (**d**) Cross-section diagram showing the assembled total hip replacement components and the position of the disc magnet (8 × 2 mm) and magnetometer PCB.

**Figure 2 sensors-25-07280-f002:**
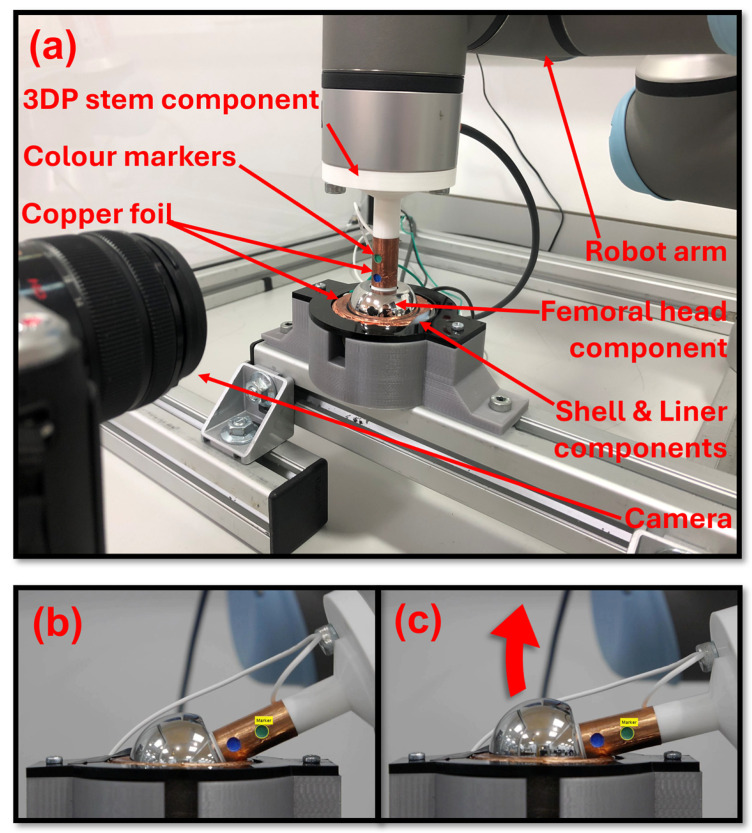
(**a**) Annotated image of the experimental test setup. (**b**) Image of the THR components recorded by the digital camera at the moment of initial component–component impingement. (**c**) Image of the THR components recorded by the digital camera at the moment of maximum bearing surface separation during a subluxation event.

**Figure 3 sensors-25-07280-f003:**
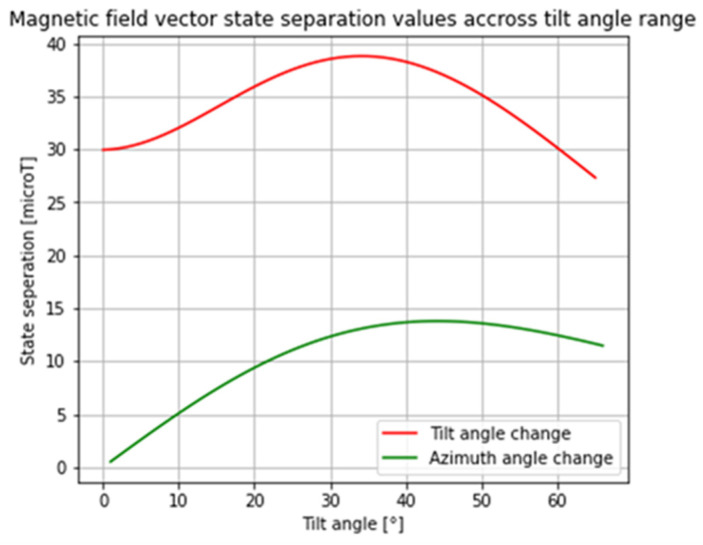
The magnetic field vector state separation when tilt state increases by 1° starting at 0° (RED) and when tilt angle is constant, and when the azimuthal angle state goes from 0° to 1° (GREEN); data shown are for the 8 *×* 2 mm disc magnet configuration.

**Figure 4 sensors-25-07280-f004:**
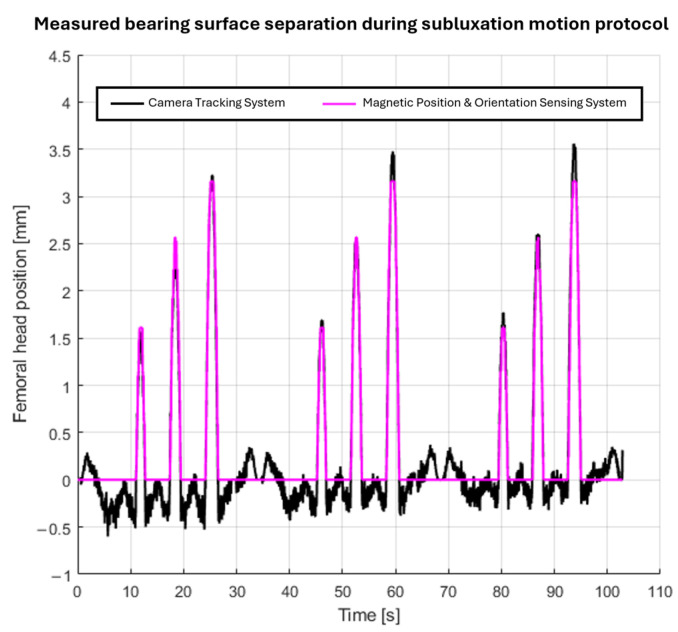
Plot of the reported centre of the femoral head position during the three repeats of the impingement-driven subluxation motion protocol by the camera tracking system and MPO sensing system.

**Table 1 sensors-25-07280-t001:** Summary of the nine magnets investigated and the meanSS and minSS values obtained in the simulated optimisation study, where N42 is the notation for the magnet material neodymium of grade 42, and Br (BRx, BRy, BRz) denotes the material remanence or magnetisation vector.

Magnet Composition	Mean State Separation (µT)	Minimum State Separation (µT)
Disc magnet 8 × 2 mm N42, Br (0,0,1300) µT	10.4	0.522
Disc magnet 8 × 1 mm N42, Br (1300,0,0) µT	4.39	0.032
Disc magnet (hyp) 8 × 2 mm N42, Br (1300,0,0) µT	8.79	0.040
Rectangle magnet 6 × 4 × 2 mm N45 H, Br (0,0,1340) µT	5.19	0.267
Rectangle magnet 2.5 × 7 × 2.5 mm N42, Br (0,0,1300) µT	4.61	0.251
Rectangle magnet (DI) 2.5 × 7 × 2.5 mm N42, Br (1300,0,0) µT	3.79	0.014
Rectangle magnet 10 × 3 × 2 mm N42, Br (0,0,1300) µT	6.13	0.281
Rectangle magnet (hyp) 7 × 7 × 2 mm N42, Br (0,0,1300) µT	10.1	0.508
Rectangle magnet (hyp) 7 × 7 × 2 mm N42, Br (1300,0,0) µT	8.57	0.044

**Table 2 sensors-25-07280-t002:** Root mean squared error, standard deviation, maximum, and minimum for actual robot stem angle versus tracking-method-predicted angle across all repeats (n = 3) for orientation tracking to two DoF motion study.

Tracking Method	RMSE Tilt Angle ± SD [°] (n = 3)	RMSE Azimuthal Angle ± SD [°] (n = 3)	Min to Max, Range Tilt Angle Error [°] (n = 3)	Min to Max, Range Azimuthal Angle Error [°] (n = 3)
Ratio stick tracking	1.1 ± 0.9	3.9 ± 3.9	−2.5 to 2.6, 5.2	−25.2 to 31.6, 56.8
Look-up table	1.0 ± 0.8	4.0 ± 4.0	−2.6 to 4.4, 7.0	−26.1 to 29.7, 55.8
Neural network	1.5 ± 1.2	6.3 ± 6.2	−2.2 to 7.6, 9.8	−85.9 to 51.6, 137.5

## Data Availability

The original contributions presented in this study are included in the article. Further inquiries can be directed to the corresponding author.

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
