# Peer review of "Measuring Relative Component Motion and Stability in Total Hip Replacements Using a Magnetic Position and Orientation Sensing System†"

_sensors, 2025, doi:10.3390/s25237280_

Round 1
Reviewer 1 Report
Comments and Suggestions for Authors
This manuscript presents an innovative and well-motivated proof-of-concept magnetic position and orientation sensing system for total hip replacement components. The work is clearly within the scope of Sensors issue. At the same time, I have the following comments that I believe, if addressed, could further strengthen the manuscript.
- At present, the captions are written in the form “Figure 1 - …”, “Table 1 - …”. According to MDPI style, captions should use the format “Figure 1. …” and “Table 1. …”, with a full stop after the number rather than a hyphen. I recommend revising all figure and table captions to follow this convention, and ensuring consistent use of “Figure 1a”, “Figure 1b”, etc. for subfigures.
- There are broken cross-references in the text (e.g. “Error! Reference source not found.”). The specific location is in the subsequent paragraphs of Table 1.
- In the Introduction, “total hip replacement (THR)” is defined and then repeated in full in the very next sentence (“over 100,000 total hip replacement (THR) surgeries…”). For conciseness and readability, it would be preferable to use the abbreviation “THR” in the second sentence once the term has been defined. Similarly, in Section 2.1 (“Materials and Methods”), the opening sentence uses the full term “A total hip replacement operates as…”. After the initial definition of “total hip replacement (THR)”, I recommend using “THR” consistently throughout the manuscript, except where the full term is needed for clarity.
- The neural network approach is only briefly described and currently performs worse than the look-up table and responsivity tracking methods (e.g., Table 2). Since all three tracking methods are presented as being investigated within the same framework, I suggest that the authors either provide more detail and discussion to explain its limited performance, or explicitly present it as a preliminary, secondary method and de-emphasise it relative to the other two.
Author Response
Comment 1: At present, the captions are written in the form “Figure 1 - …”, “Table 1 - …”. According to MDPI style, captions should use the format “Figure 1. …” and “Table 1. …”, with a full stop after the number rather than a hyphen. I recommend revising all figure and table captions to follow this convention, and ensuring consistent use of “Figure 1a”, “Figure 1b”, etc. for subfigures.
Response 1:
Edits made as tracked changes in MS word.
Comment 2: There are broken cross-references in the text (e.g. “Error! Reference source not found.”). The specific location is in the subsequent paragraphs of Table 1.
Response 2:
Edit made as a tracked change in MS word, “Error! Reference source not found.” Replaced with correct reference to Figure 3.
Comment 3: In the Introduction, “total hip replacement (THR)” is defined and then repeated in full in the very next sentence (“over 100,000 total hip replacement (THR) surgeries…”). For conciseness and readability, it would be preferable to use the abbreviation “THR” in the second sentence once the term has been defined. Similarly, in Section 2.1 (“Materials and Methods”), the opening sentence uses the full term “A total hip replacement operates as…”. After the initial definition of “total hip replacement (THR)”, I recommend using “THR” consistently throughout the manuscript, except where the full term is needed for clarity.
Response 3:
Edits made as tracked changes in MS word. I have now edited all incidences in the main body of text to THR, where “total hip replacement” is mentioned in headings or captions I have left it un-abbreviated and in the discussion where clarity was needed when comparing to a “total knee replacement” I have left it un-abbreviated.
Comment4: The neural network approach is only briefly described and currently performs worse than the look-up table and responsivity tracking methods (e.g., Table 2). Since all three tracking methods are presented as being investigated within the same framework, I suggest that the authors either provide more detail and discussion to explain its limited performance, or explicitly present it as a preliminary, secondary method and de-emphasise it relative to the other two.
Response 4:
Thank you for the fair comment, I have now included more detail and discussion as to why the neural network method may have underperformed when compared to the other tracking methods. I have included the following in the 4th paragraph of the discussion:
“The comparatively poor performance of the neural network method suggests the model is not fully optimised. The training dataset size and the number and size of network layer were not altered and there was no examination into whether the model was over or under fitted. Future work can explore addressing these factors with the aim of improving the performance of the Neural Network thus reducing the error of the orientation prediction.”
Reviewer 2 Report
Comments and Suggestions for Authors
The manuscript presents a technically ambitious proof-of-concept system for tracking relative motion in total hip replacement components using a magnet-sensor configuration. Its originality lies primarily in extending magnetic position and orientation (MPO) sensing, previously used in joystick and knee prosthesis contexts, for a far more constrained geometric and clinical environment of a total hip replacement. This focus is novel because THR sensorisation has traditionally centred on force measurement or loosening detection, and the authors correctly position their work as the first to apply MPO technology to continuous kinematic monitoring in hip implants. The rationale is well defined and supported by prior literature summarized in the introduction, and the system demonstrates meaningful clinical potential through its ability to detect femoral head separation during impingement-driven subluxation events. The work is therefore conceptually original and addresses a relevant gap in orthopaedic sensing technologies. 
The methodology is detailed and generally sound, with clear descriptions of device integration, magnet optimisation, calibration, simulation work, and validation procedures. The motion routines using a UR3 robotic arm provide a repeatable way to generate good data, and the addition of a camera tracking system for subluxation validation strengthens the experimental framework. The manuscript benefits from explicit reporting of sampling rates, calibration methods, and data-synchronisation steps, which enhances transparency and makes the methodology easier to reproduce. 
Some methodological limitations, however, reduce the robustness of the conclusions. The sensing performance for azimuth angle at low tilt (<20°) is affected by the sensor resolution relative to the small state separation between adjacent states, an issue the optimisation study already highlights. This limitation leads to elevated azimuthal errors and becomes a structural constraint of the current 2 DoF formulation. The authors acknowledge this but do not attempt an intermediate mitigation strategy, such as adaptive state spacing or noise-aware modelling. The neural-network tracking model also performs substantially worse than the look-up-table and responsivity methods, suggesting that either the network architecture or the training dataset was insufficiently tuned. While the authors frame this as an exploratory comparison, the paper would benefit from deeper reflection on why the neural network underperformed, especially given prior work showing good performance in related MPO applications. Another methodological constraint is that the subluxation simulation does not apply anatomically realistic axial loading, which the authors correctly note would influence femoral head displacement patterns, meaning that the reported accuracy of 0.2 mm RMSE should be interpreted with caution when extrapolating to physiological conditions.
I would suggest commenting regard the osseointegration of the prosthesis and the long-term evolution of the bone tissue surrounding the implant (see e.g., [1,2]).
[1] Allena, R., Scerrato, D., Bersani, A., & Giorgio, I. (2025). Simulating bone healing with bio-resorbable scaffolds in a three-dimensional system: insights into graft resorption and integration. Comptes Rendus. Mécanique, 353(G1), 479-497.
[2] Lekszycki, T., & dell'Isola, F. (2012). A mixture model with evolving mass densities for describing synthesis and resorption phenomena in bones reconstructed with bio‐resorbable materials. ZAMM‐Zeitschrift für Angewandte Mathematik und Mechanik, 92(6), 426-444.
The mathematical expressions used throughout the paper are mostly correct. 
Overall, the manuscript presents an inventive and promising approach to in-vivo kinematic monitoring in total hip replacements. Its originality is clear, the methodology is thoughtfully constructed, and the results demonstrate encouraging accuracy for a first-generation prototype. However, limitations in azimuthal resolution at low tilt and the simplified biomechanics of the subluxation model suggest that the work, while strong as a proof of concept, would benefit from further refinement before clinical translation. The study lays a solid foundation for future development, especially toward three-degree-of-freedom tracking and more physiologically realistic loading scenarios.
Author Response
Comment1:
While the authors frame this as an exploratory comparison, the paper would benefit from deeper reflection on why the neural network underperformed, especially given prior work showing good performance in related MPO applications.
Response 1:
Thank you for the fair comment, I have now included more detail and discussion as to why the neural network method may have underperformed when compared to the other tracking methods. I have included the following in the 4th paragraph of the discussion:
“The comparatively poor performance of the neural network method suggests the model is not fully optimised. The training dataset size and the number and size of network layer were not altered and there was no examination into whether the model was over or under fitted. Future work can explore addressing these factors with the aim of improving the performance of the Neural Network thus reducing the error of the orientation prediction.”
Comment 2:
Another methodological constraint is that the subluxation simulation does not apply anatomically realistic axial loading, which the authors correctly note would influence femoral head displacement patterns, meaning that the reported accuracy of 0.2 mm RMSE should be interpreted with caution when extrapolating to physiological conditions.
I would suggest commenting regard the osseointegration of the prosthesis and the long-term evolution of the bone tissue surrounding the implant (see e.g., [1,2]).
[1] Allena, R., Scerrato, D., Bersani, A., & Giorgio, I. (2025). Simulating bone healing with bio-resorbable scaffolds in a three-dimensional system: insights into graft resorption and integration. Comptes Rendus. Mécanique, 353(G1), 479-497.
[2] Lekszycki, T., & dell'Isola, F. (2012). A mixture model with evolving mass densities for describing synthesis and resorption phenomena in bones reconstructed with bio‐resorbable materials. ZAMM‐Zeitschrift für Angewandte Mathematik und Mechanik, 92(6), 426-444.
Response 2:
Thank you for the constructive comment I have now included further comments on alternate methods to simulate a more biomechanically accurate impingement driven subluxation event. I have also included considerations of how physiological conditions will affect the long term positions of the components and how this would impact the error of the system. I have included the following at the end of the 5th paragraph of the discussion:
“This will likely be achieved by using validated in-vitro experimental hip simulators [27] or using a more capable robotic arm like that used by [28,29]. Furthermore, when consid-ering the physiological environment, the system will need to account for long term changes in the position of the components. For example, because of implant precession, osseointegration of the prosthesis and the evolution of the bone material around the im-plant, which will affect the achievable accuracy of the system.”